# Physical Properties of Paste Synthesized from Wet- and Dry-Processed Silver Powders

**DOI:** 10.3390/ma17061273

**Published:** 2024-03-09

**Authors:** Hyun Jin Nam, Minkyung Shin, Hye Young Koo, Se-Hoon Park, Hyun Min Nam, Su-Yong Nam

**Affiliations:** 1ICT Device Packaging Research Center, Korea Electronics Technology Institute, Seongnam 13509, Republic of Korea; hjnam1203@keti.re.kr (H.J.N.);; 2Electronic Convergence Materials and Devices Research Center, Korea Electronics Technology Institute, Seongnam 13509, Republic of Korea; 3Powder Materials Division, Korea Institute of Materials Science, Changwon 51508, Republic of Korea; hykoo@kims.re.kr; 4FP Co., Ltd., Busan 47047, Republic of Korea; 5Department of Nanotechnology Engineering, Pukyong National University, Busan 48513, Republic of Korea

**Keywords:** conducting powder, dry synthesis, silver paste, rheology, screen printing

## Abstract

This study compares the characteristics and low-temperature curing properties of pastes prepared from silver (Ag) powders synthesized by either wet powder (WP) or dry powder (DP) processing. The WP synthesis of electrode particles has the advantage of controlling the average particle size and particle size distribution but the disadvantage of producing low-purity, crystalline particles because they are synthesized through chemical reduction at less than 100 °C. Conversely, the DP synthesis of electrode particles has the advantage of producing pure, highly crystalline particles (due to synthesis at high temperatures) but the disadvantage of a high processing cost. WP and DP were used to manufacture pastes for low-temperature curing, and the physical properties of the pastes and the electrode characteristics after low-temperature curing were compared between powder types. Shear stress as a function of the shear rate shows that the WP paste is a plastic fluid, whereas the DP paste is a pseudoplastic fluid, closer to a Newtonian fluid. Screen printing the Ag pastes and curing for 30 min at 130 °C produces a nonconductive WP paste, whereas it produces a DP paste with a conductivity of 61 mΩ/sq, indicating that the highly crystalline DP paste is advantageous for low-temperature curing.

## 1. Introduction

Ag has been widely utilized in the fabrication of electrical contacts for solar cells, hybrid circuits, and various other devices owing to its exceptional electrical properties [1,2,3,4]. The synthesis of Ag powder, particularly particles smaller than 1 μm, has primarily relied on liquid-based processing methods because the ductility of Ag makes it difficult to produce fine particles through pulverization [5,6,7,8,9,10,11]. While liquid processing methods facilitate the large-scale synthesis of particles with a desired average size and narrow size, controlling the carbon content of particles to less than 1 wt% remains a challenge because of the use of chemical reducing agents. In addition, synthesis below 100 °C produces particles with low crystallinity [12,13].

In the case of Ag applied to high-temperature sintering, carbon is removed through a high-temperature heat treatment process and crystals grow at the same time, so low carbon and high crystalline characteristics of the raw material powder are not necessarily required. However, flexible and stretchable materials, which have recently become increasingly in demand, require highly pure and crystalline raw materials because post-heat treatment is performed below 200 °C to prevent the hardening of the material [14,15,16].

To obtain highly pure and crystalline particles smaller than 1 μm, we synthesized Ag powder through a dry synthesis method based on a spray pyrolysis process. In spray pyrolysis, fine droplets generated by a liquid droplet generator are carried through a high-temperature furnace (>900 °C) by a carrier gas and thereby dried and pyrolyzed, producing a powder. This process is advantageous because it produces highly pure and crystalline metal or oxide particles depending on the carrier gas [17,18,19,20].

The Ag particle paste can be classified as hardened Ag paste or high-temperature-sintered Ag paste. Low-temperature-hardened Ag paste has a curing temperature of 100 °C–200 °C and is mainly used as electrodes for flexible or stretchable substrates such as polyethylene terephthalate (PET). The binder used to manufacture the paste remains intact, and the Ag particles stick together (contact) and conduct [21,22,23]. On the other hand, high-temperature-sintered Ag paste, which is mainly used on substrates such as wafers or tempered glass, achieves conductivity by sintering Ag particles after thermally decomposing the binder at high temperatures. The resistivity of low-temperature-hardened Ag is 10^−4^–10^−5^ Ω cm after curing [24,25], whereas the resistivity of high-temperature-sintered Ag paste is about 10^−6^ Ω cm after sintering [26,27].

Screen printing is the most used electronics printing technology to form electrodes. Screen printing technology requires an Ag paste with specific rheological characteristics because the results and conductivity depend strongly on the rheological characteristics of Ag paste [28,29,30,31,32,33].

This study evaluated the particle size distribution and x-ray diffraction (XRD) of Ag particles synthesized by different methods. These particles were then used to form, via low-temperature curing, electrodes with high conductivity. We also evaluated the rheological characteristics of paste manufactured from Ag particles synthesized by the WP and DP processes. We confirmed the thixotropy characteristics based on shear stress as a function of the shear rate, the viscosity characteristics as a function of the shear rate, the storage modulus G’, the loss modulus G’’, and the tan δ characteristics as functions of frequency. Based on G’ and G′′ as functions of shear stress, we evaluated the three-dimensional (3D) network structure formed by the Ag particles in the paste. Finally, by using scanning electron microscopy (SEM), we observed the surface smoothness of the cured electrode and the packing density of the Ag particles after printing Ag paste on PET. We also measured the electrical conductivity of the resulting electrodes to determine the material characteristics for the different synthesis methods.

## 2. Materials and Methods

### 2.1. Synthesis Method for Ag Powder

For this study, Ag paste was manufactured using Ag powder synthesized by a wet process (HP0704, LT metal Co., Ltd., Incheon, Republic of Korea) and DP synthesized by a spray pyrolysis process. The spray pyrolysis process involves generating a micrometer-sized droplet of solution containing the desired metal salt using an ultrasonic nebulizer. This droplet is continuously synthesized into submicron-sized particles within a few seconds by passing it through a high-temperature furnace using a carrier gas by heat and a gas atmosphere. In this study, the process involves rapid drying, thermal decomposition, and a melting process within a short time, less than 3 s, passing through a furnace with a temperature exceeding 900 °C. These results indicate high crystallinity, low resistance, and enable the synthesis of particles with high purity without the need for a chemical reducing agent.

The morphology, average particle size, and particle size distribution of the synthesized powders were determined by field emission SEM (JSM-5800, JEOL Ltd., Akishima, Japan). The specific surface area was measured using a Brunauer–Emmett–Teller (BET) instrument (BELSORP-max, MicrotracBEL Co., Osaka, Japan), and the pretreatment conditions were maintained at 100 °C for 16 h to prevent Ag sintering due to the heat of the pretreatment. For crystallographic comparison, the phase was analyzed using x-ray diffraction (XRD) (D/Max 2500, RIGAKU Co., Tokyo, Japan), and the crystal size was calculated using the Scherrer equation. In addition, carbon analysis was conducted using a carbon–sulfur analyzer (CS 744, LECO Co., Michigan, United States of America) to confirm the carbon in the powder.

### 2.2. Preparation of Ag Paste

Ag paste was manufactured using two different types of particles (see Table 1). Figure 1 shows the molecular structure of the binder used to disperse the Ag powder. The acrylic (Z251, Daicel-allnex Ltd., Tokyo, Japan) binder was diluted (N.V. = 45%) in dipropylene glycol monomethyl ether solvent (DUKSAN Co., Ltd., Seoul, Republic of Korea), which has a molecular weight of 15,000, a double bond equivalent of 380, and an acid value (KOHmg/g) of 66. It uses an acrylic polymer resin with a Tg of 136 °C. This binder is halogen-free, heat-resistant, weather-resistant, and can serve as a low-temperature curing material because of its ability to cure under ultraviolet, electron-beam, and thermal conditions.

Figure 2 shows the process for manufacturing these Ag pastes. The dispersant BYK-180 (www.byk.com) was dissolved in the solvent carbitol acetate (Youngshin Chemical Co., Republic of Korea) and then added to the binder. A small amount of Ag powder was added to this binder and mixed first using a premixing paste mixer (PDM-300, Daewha Tech Co., Ltd., Gimhae, Republic of Korea). The Ag paste was mixed for a second time using TRM (TRM-6.5, Kyung Yong Machinery Co., Ltd., Ansan, Republic of Korea) to disperse the Ag particles more uniformly.

### 2.3. Evaluation of Rheology and Conductivity

The rheological properties of the manufactured Ag paste were evaluated at room temperature using a rheometer (HAAKE Rheostress1, Thermo Fisher Scientific Inc., Federal Republic of Germany). To examine the flow characteristics of the Ag paste, thixotropy, viscosity, elasticity, frequency sweep, and elasticity were measured as functions of frequency. To determine the conductivity, the Ag paste was printed using a screen plate made of stainless 500 mesh on PET (125 µm) and a semiautomatic screen printer (FORCE 2525, Minogroup, Gifu, Japan). The printed electrode film was cured in an oven dryer at 130 °C for 30 min, following which the conductivity was measured using a four-point probe (DMM6500, Keithley Inc., Ohio, United States of America).

### 2.4. Evaluation of Surface

After screen printing the Ag paste and curing it in an oven dryer at 130 °C for 30 min, the smoothness of the surface was photographed through a microscope set at various magnifications. The electrode surface was photographed using an optical microscope, and the roughness was confirmed using a 3D microscope (LSM 800 MAT, Carl Zeiss Co., Ltd., Oberkochen, Germany). Finally, the surface and cross-sectional shape of the cured electrode film were imaged using an SEM (VEGA II LSU, Tescan Korea, Ltd., Seoul, Republic of Korea).

## 3. Results and Discussion

### 3.1. Characteristics of Ag Particles

This study used WP Ag and DP Ag to fabricate pastes, which in turn were used to fabricate electrodes. Before fabricating the electrodes, the shape, purity, and crystallinity of the WP and DP were compared to determine their individual characteristics. Figure 3 shows SEM images and the particle size distribution of the Ag powders. Both powders have spherical particles, and the average particle size is 360 and 480 nm for the WP and DP, respectively. Figure 3c shows that the DP has a wider particle size distribution than the WP.

To confirm the porosity of the WP and DP, we conducted a specific surface area analysis by BET. Figure 4 show the N_2_ adsorption–desorption isotherms graphs of the DP and WP. To prevent the characteristics of the particles from changing because of sintering during pretreatment at 200 °C or higher, pretreatment was performed at 100 °C for 16 h. Through the graph in Figure 4, both WP and DP can be considered dense particles. The specific surface area of the WP and DP was 1.7 and 0.86 m^2^/g, respectively. Assuming particles are (1) completely spherical, (2) have no internal pores, and (3) have a uniform particle size distribution, the relationship between the spherical surface area and diameter can be described by Equation (1). Utilizing this equation, the mean particle size can be estimated from the given specific surface area (S) values. For Ag with a density (ρ) of 10.49 g/cm^3^, the calculated mean particle sizes (d) of the WP and DP were 336 and 665 nm, respectively.
D = 6000/(ρ × S)(1)

The residual carbon of the particles determined using a carbon–sulfur analyzer gives 0.3 and 0.01 wt% for WP and DP, respectively, with WP showing higher carbon values [34].

Figure 5 shows the XRD results for the two powders. To compare the crystallinity of the WP synthesized through chemical reduction and DP synthesized through a high-temperature melting process, XRD analysis was measured. Using the Scherrer equation, the crystallite sizes were measured, revealing a size of 31 nm for WP and 74 nm for DP. The crystallite size of the DP was approximately 2.4 times larger than that of the WP.

### 3.2. Rheological Characteristics and Screen Printing of Ag Paste

Figure 6 shows photographs of the Ag paste produced from WP and DP Ag. Due to the different average particle sizes, the WP paste is dark gray, whereas the DP paste is light gray.

The thixotropy of the Ag pastes was evaluated by measuring shear stress as a function of the shear rate (see Figure 7). Thixotropy is a physical measurement that gives information on the network structure formed by dispersed particles. Shear stress versus the shear rate of the acrylic binder solution was first measured and reveals weak hydrogen bonds in the polymer but no thixotropic loop. Pasting this binder with DP Ag produces a negligible thixotropic loop in the form of a pseudo plastic fluid (similar to a Newtonian fluid) due to the Ag particles of the powder. Pasting this binder with WP Ag produces a clear plastic fluid (viscoplastic), showing that shear stress increases considerably and nonlinearly as the shear rate increases. When the shear rate decreases, the linear form of shear stress decreases, which is attributed to the structure destroyed by strong shear stress gradually recovering to its original structure. The 3D network structure formed by Ag particles is explained in detail below [35,36,37].

Next, Figure 8a shows the results of frequency sweep measurements, which are used to characterize the structure formed by the WP and DP Ag. For the acrylic binder solution, the storage modulus G′ is less than the loss modulus G” even at a low angular frequency, like a liquid particle suspension. This behavior is characteristic of a suspension in which particles are dispersed in a liquid. For DP Ag in this binder solution, the storage modulus and loss modulus increase monotonically. At low (high) angular frequency, G″ is less (greater) than G′. This phenomenon is attributed to the destruction of the weak 3D base network structure at a high frequency [38,39].

The difference between the storage modulus and the loss modulus appears clearly over the entire frequency range when the WP Ag is mixed and pasted. This result is indicative of a dense network structure formed by the WP Ag particles. This phenomenon also occurs in gel samples that form a 3D network structure. Figure 8b shows tan δ as a function of angular frequency. For the acrylic binder, tan δ decreases as the angular frequency increases, whereas it increases slightly after going through a minimum for the Ag pastes. The DP paste produces a larger tan δ than the WP paste. Generally, tan δ > 3 indicates no aggregation (strong bonding), and 1 < tan δ < 3 indicates a weakly bonded paste. Tan δ < 1 indicates a paste close to a gel state that forms a strong 3D network structure [40].

To examine the flow characteristics of the Ag paste, we measured the viscosity as a function of the shear rate [see Figure 9a]. First, the acrylic binder solution exhibits a Newtonian flow (i.e., almost no change in viscosity even when the shear rate increases). The viscosity of the DP paste in this binder gradually decreases as the shear rate increases. Thus, the DP paste in the binder produces a pseudoplastic behavior with excellent flowability. This phenomenon is attributed to a weak binding structure formed by the Ag particles. That is, this paste has negligible thixotropy. The thixotropy index (TI value: 5/50 s^−1^) of this paste is approximately 1.25 [41,42].

However, for WP paste in the binder solution, the viscosity decreases sharply with an increasing shear rate (plastic flow). This result is attributed to the dense 3D network structure formed by the Ag particles. In other words, this paste has strong thixotropy, as indicated by the thixotropy index of approximately 2.33, which is significantly greater that of the DP paste.

Finally, to understand the storage stability and critical stress of the Ag paste, Figure 9b shows the measured storage and loss moduli as functions of shear stress. The DP paste maintains a stable 3D network up to approximately 4 Pa of shear stress. Flow starts at approximately 7 Pa (G′ < G″). This means that the 3D network structure is destroyed by even a small shear stress, indicating that the DP Ag particles form a weak 3D network structure.

The WP paste produces a stable 3D network structure up to approximately 30 Pa. However, flow occurs at shear stress >50 Pa (G′ < G″). This means that strong shear stress is required to destroy the 3D network structure. The WP paste with its strong 3D network structure is more stable than the DP paste.

Figure 10 shows schematically the structure of the Ag pastes. Figure 10a,b show the 3D network structure of the DP and WP pastes. The DP paste forms a structure that succumbs to fluidity under small shear stress because the Ag particles form a loose 3D network structure. In contrast, the WP paste starts to flow only upon applying a strong shear stress (>50 Pa) because the Ag particles form a dense 3D network structure.

Figure 11 shows the results obtained by screen printing (500-mesh plate) Ag paste with the aforementioned rheological characteristics. Optical microscope photographs are shown immediately after printing (before curing) and after curing at 130 °C for 30 min. The DP paste forms a smooth film before and after curing, but in the case of the WP paste, the surface shows a somewhat rough film. This result indicates good leveling, which is attributed to the excellent recovery characteristics of the paste that passes through the screen mesh. This result is attributed to a 3D network structure that is easily destroyed and becomes fluid even under small shear stress [43,44,45].

The WP paste forms a dense 3D network structure, so it is less smooth than the DP paste. This result is attributed to the stronger shear stress required for fluidity.

Figure 12 shows the results of screen printing a small pattern using DP and WP pastes (cured at 130 °C for 30 min). The roughness of the edges of the printed patterns differs significantly, which is attributed to the same flow characteristics discussed for screen printing (Figure 10). Therefore, DP paste is better suited for applications requiring smooth pattern film and sharp patterns. The surface roughness of the printed electrodes was evaluated using 3D laser (confocal) equipment. The Ra (arithmetic mean height) value, which indicates the average roughness, was measured to be 0.08 µm for the DP paste and 0.28 µm for the WP paste. It was confirmed that the surface roughness was higher for the WP paste.

### 3.3. Electrical Characteristics of Cured Electrode Film

Three sheets of DP paste and three of WP paste were screen-printed and then cured at 130 °C for 30 min. Figure 13 shows photographs of the conductivity measurements of these sheets. The DP paste produces sheets with good conductivity, whereas the WP paste produces nonconductive sheets. This result is attributed to the difference in carbon residue remaining on the surface of the Ag particles. The carbon content remaining on the surface of the DP and WP Ag is 0.01 and 0.3 wt%, respectively. In other words, the WP paste produces nonconducting sheets because significantly more carbon remains on the surface of the Ag particles than is the case for DP paste.

Figure 13a shows the results of surface resistivity measurements made at five locations on the DP electrode surface. The average surface resistivity is 61 m/sq.

Figure 14 shows the surface and cross-sectional SEM images of cured electrode films made from WP (Figure 14a) and DP (Figure 14b) pastes. The SEM results show that both types of Ag pastes form films of high packing density. The SEM images reveal no significant difference between the WP paste and DP paste for cured electrode films.

## 4. Conclusions

In this study, we examined the influence of particles synthesized through different methods on low-temperature sintering materials, focusing on high-purity and high-crystallinity particles. Wet-processed particles with low crystallinity below 3N grade, synthesized at temperatures below 100 °C, were compared with physically synthesized high-crystallinity 4N-grade particles obtained at temperatures exceeding 900 °C. Pastes were prepared using these particles, and the variations in the paste properties based on the synthesis methods were examined. The resulting differences in electrode characteristics were also compared.

The rheological characteristics, screen printability, and electrical conductivity of the WP and DP Ag pastes were all measured. The results lead to the following conclusions:

The specific surface areas of WP and DP are 1.70 and 0.86 m^2^/g, respectively. This result is confirmed numerically and by SEM images, which show that the WP particles are smaller. The residual carbon in the particles determined by using a carbon–sulfur analyzer is 0.3 wt% and 0.01 wt% for WP and DP, respectively, indicating that the WP contains more carbon.

Pastes were made from DP and WP Ag, and their rheological characteristics were determined. Shear stress as a function of the shear rate shows that the DP paste behaves as a pseudoplastic fluid, whereas the WP paste behaves as a plastic fluid (viscoplastic). In addition, frequency sweeps and tan δ show that the DP paste forms a loose 3D network structure, whereas the WP paste forms a gel with a dense 3D network structure. Finally, the viscosity and viscoelasticity as a function of the shear rate show that the DP paste forms a weak 3D structure, whereas the WP paste forms a considerably stronger 3D network structure.

The DP and WP pastes were screen-printed to analyze the smoothness of the resulting electrode surface. The DP paste forms smooth electrodes, whereas the WP paste forms rough electrodes. This result is attributed to the different structures formed by the Ag particles in the paste state and the different thixotropy indexes (DP paste has a TI of 1.25, and WP paste has a TI of 2.33).

After screen printing the DP and WP pastes and curing the resulting electrode film at 130 °C for 30 min, the surface resistance was measured with a four-point probe. The DP paste produces an excellent sheet resistivity of about 61 mΩ/sq, whereas the WP paste produces nonconducting sheets. This result is attributed to the greater amount of organic matter remaining on the surface of the WP Ag particles than on the DP Ag particles.

These results show that DP Ag (with high crystallinity, high purity) particles are better suited than WP Ag particles for applications involving low-temperature curing Ag pastes.

## Figures and Tables

**Figure 1 materials-17-01273-f001:**
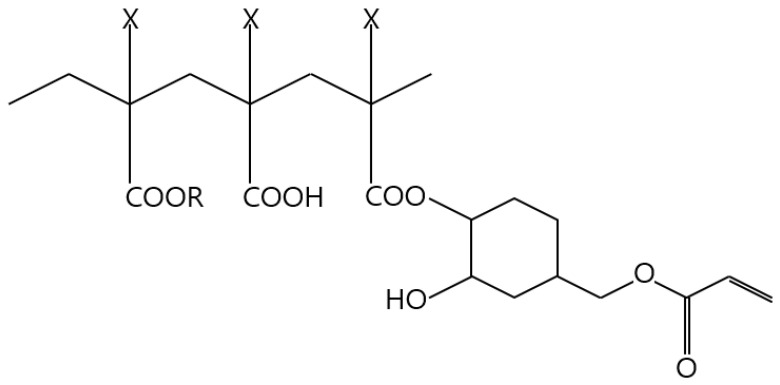
Molecular structure of the binder radical curing-type acrylic polymer.

**Figure 2 materials-17-01273-f002:**
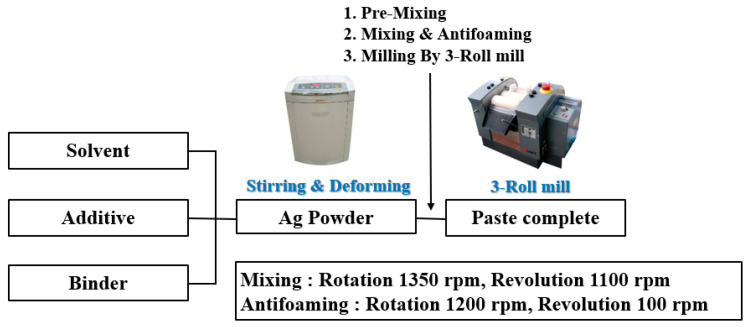
Process for fabricating nano Ag paste.

**Figure 3 materials-17-01273-f003:**
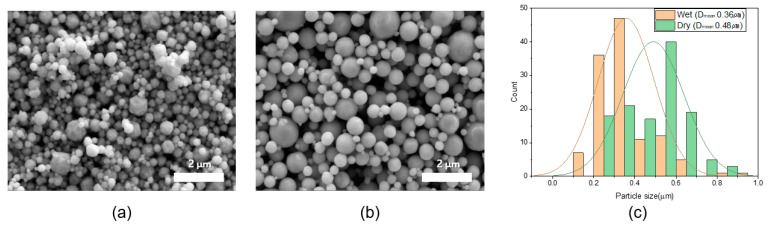
SEM image of (**a**) WP, (**b**) DP, and (**c**) particle size distribution.

**Figure 4 materials-17-01273-f004:**
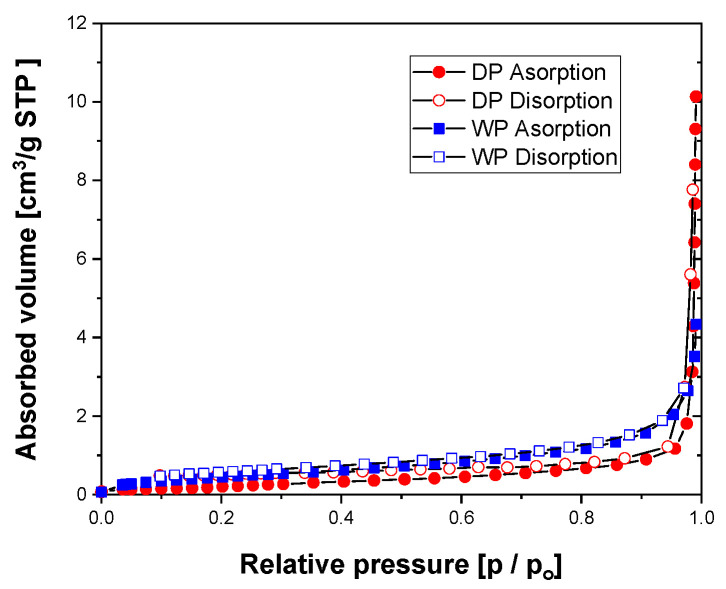
The N2 adsorption–desorption isotherms graphs of DP and WP.

**Figure 5 materials-17-01273-f005:**
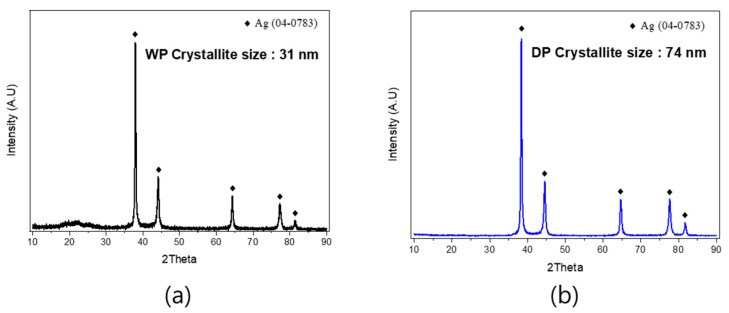
XRD results of (**a**) WP and (**b**) DP.

**Figure 6 materials-17-01273-f006:**
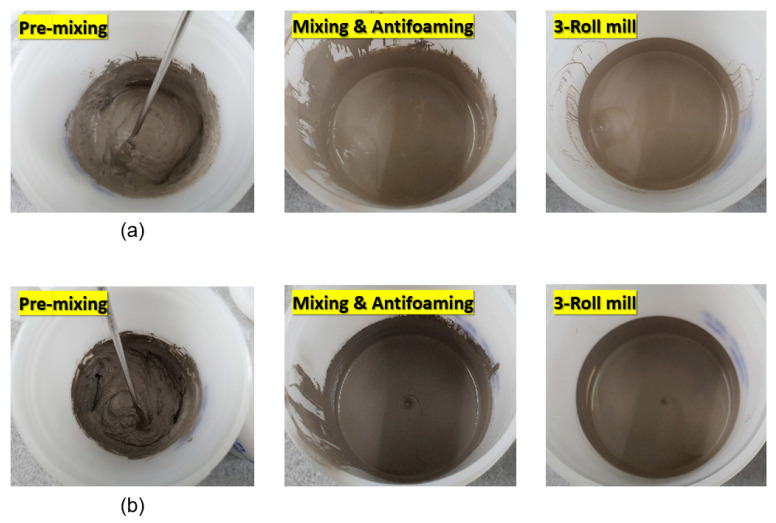
Photographs of DP paste (**a**) and WP paste (**b**) during manufacturing of Ag paste.

**Figure 7 materials-17-01273-f007:**
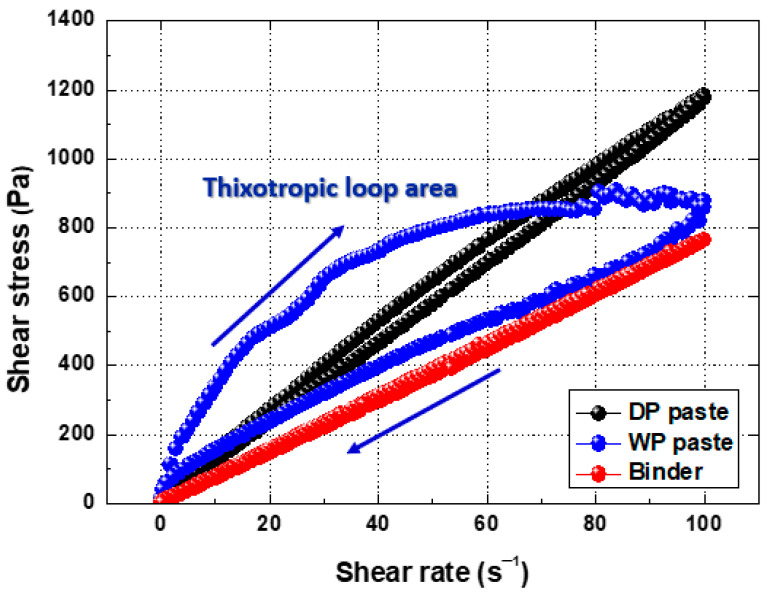
Shear stress as a function of shear rate for binder and WP and DP paste.

**Figure 8 materials-17-01273-f008:**
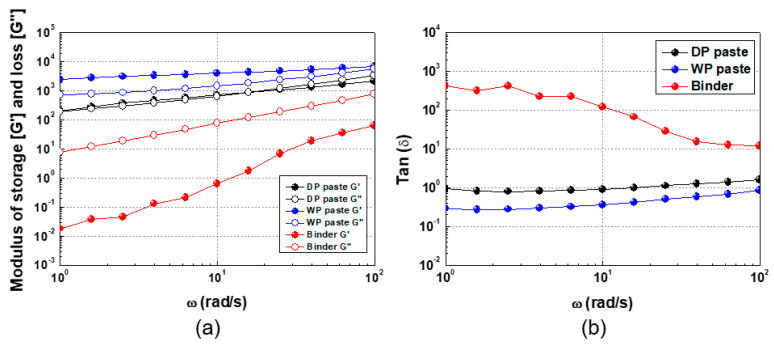
(**a**) Storage and loss modulus as a function of angular frequency and (**b**) tan δ(G″/G′) as a function of angular frequency.

**Figure 9 materials-17-01273-f009:**
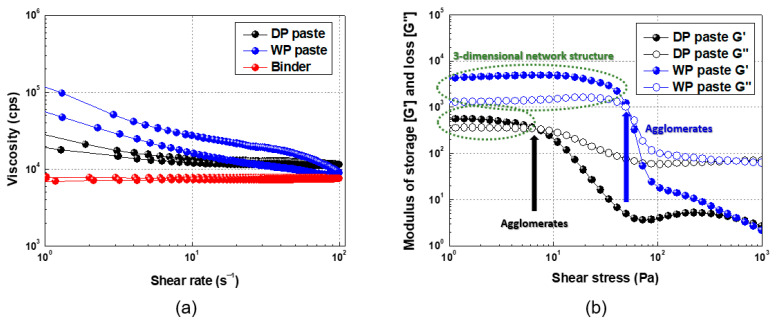
(**a**) Viscosity as a function of shear rate. (**b**) Storage and loss moduli as functions of shear stress.

**Figure 10 materials-17-01273-f010:**
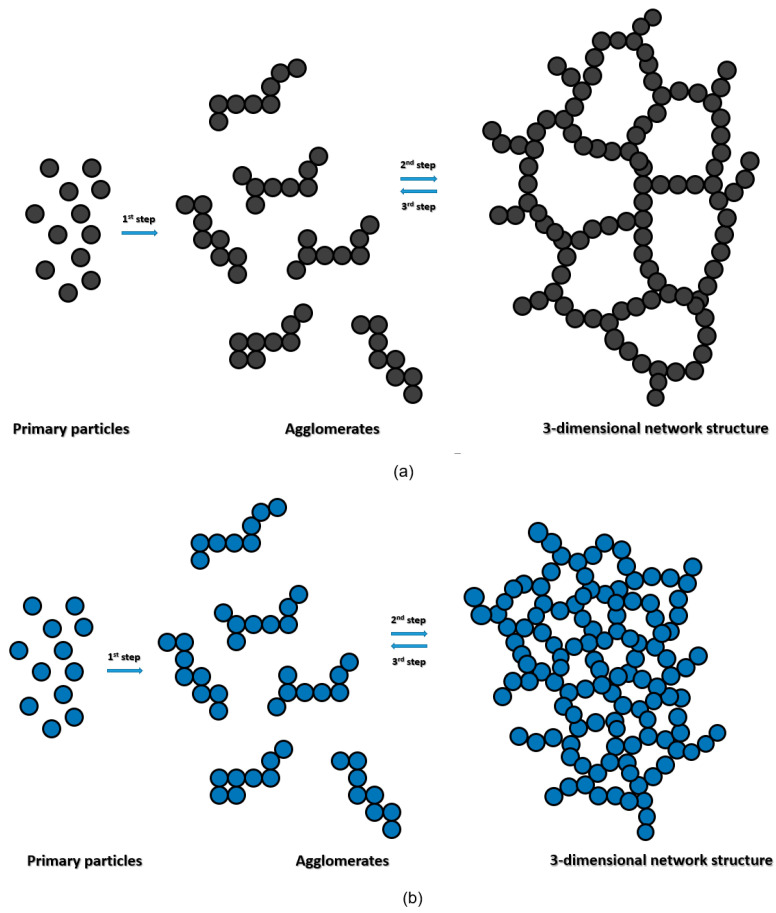
Schematic diagrams of 3D network structure of (**a**) DP paste and (**b**) WP paste.

**Figure 11 materials-17-01273-f011:**
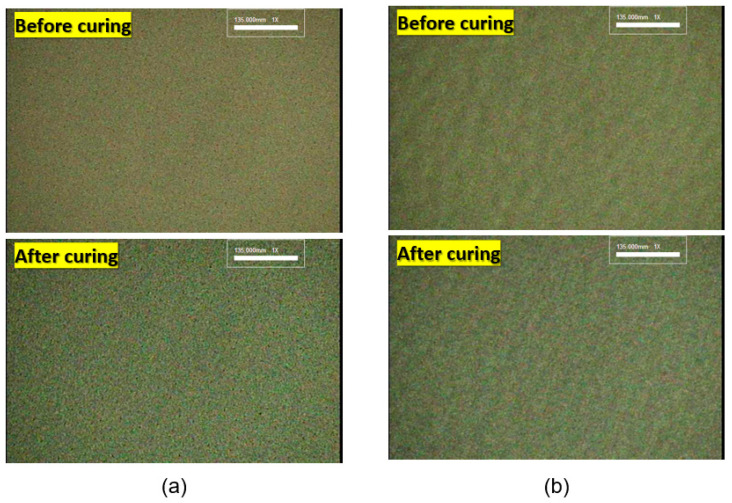
Smoothness after screen printing of (**a**) DP paste and (**b**) WP paste.

**Figure 12 materials-17-01273-f012:**
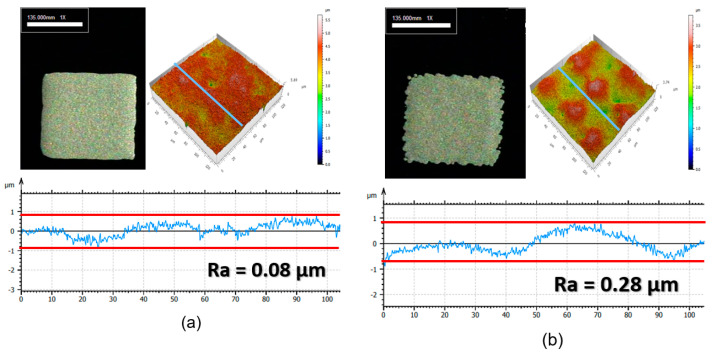
Pattern roughness after screen printing with (**a**) DP paste and (**b**) WP paste.

**Figure 13 materials-17-01273-f013:**
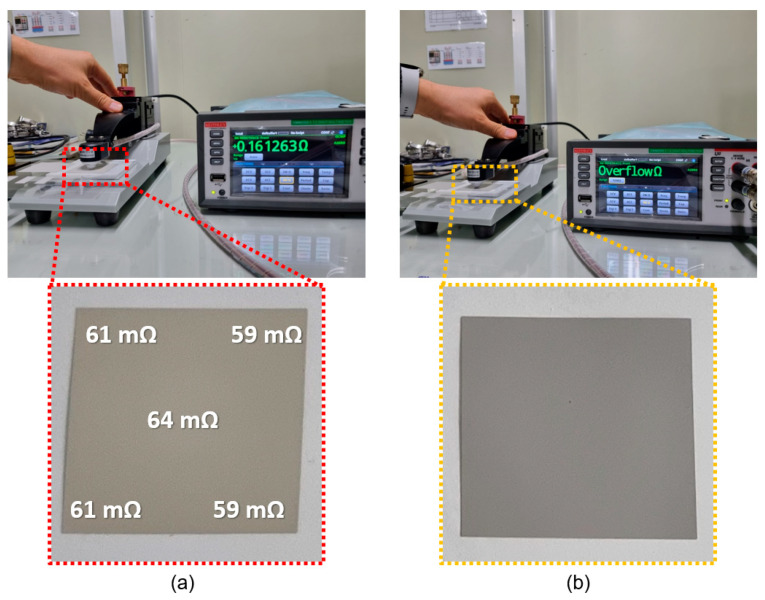
Surface resistivity results for (**a**) DP paste and (**b**) WP paste.

**Figure 14 materials-17-01273-f014:**
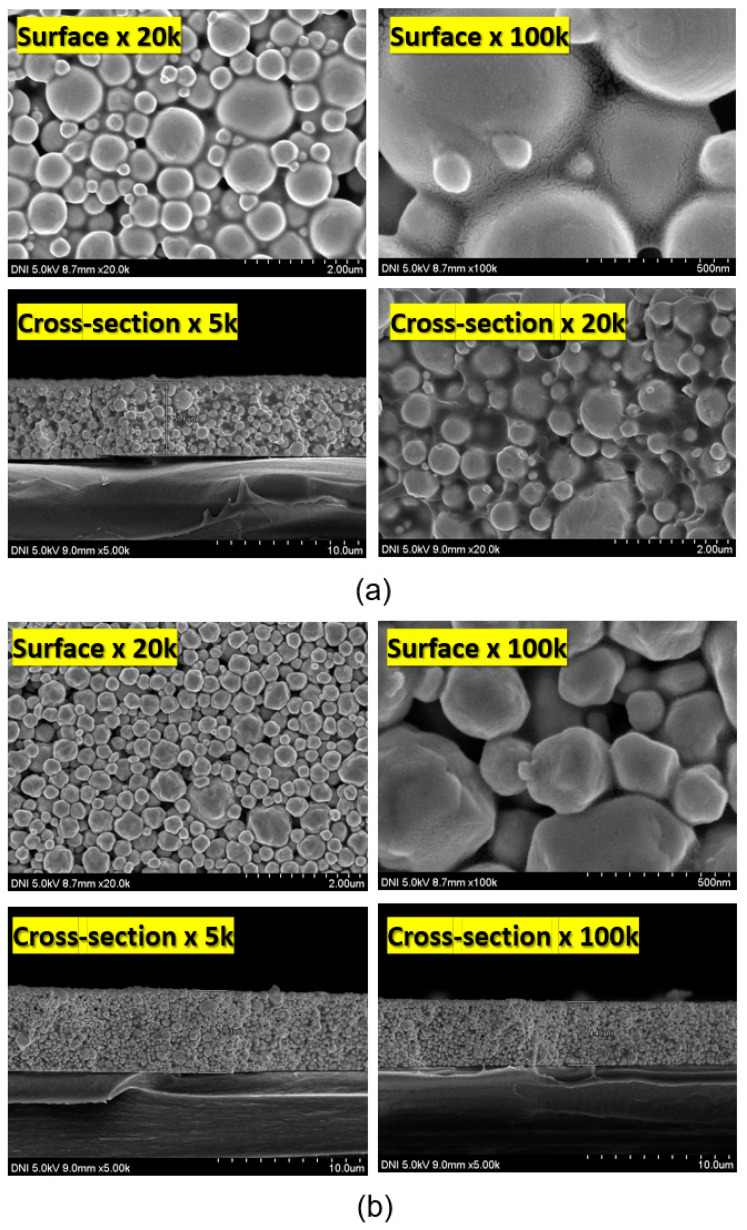
Surface and cross-sectional SEM images of (**a**) DP paste and (**b**) WP paste.

**Table 1 materials-17-01273-t001:** Formulation of WP and DP pastes.

Wt%	Filler	Binder	BYK-180	Carbitol	Total
WP	86.49%	10.81%	0.54%	2.16%	100%
DP	86.49%	10.81%	0.54%	2.16%	100%

## Data Availability

Data are contained within the article.

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
