# Peer review of "Physical Properties of Paste Synthesized from Wet- and Dry-Processed Silver Powders"

_materials, 2024, doi:10.3390/ma17061273_

Round 1

Reviewer 1 Report

Comments and Suggestions for Authors

In this paper, the authors have synthetised and compared wet and dry silver nanoparticles. These particles were used to fabricate pastes, which were characterised.

Line 38-39, revise the sentence as it doesn't make any sense.

Check the English for posttreatment.

Line 56-58, add reference.

Paragraph 2.1...have you synthetised the powders? If yes, can you detail the process?

For the reagents and equipment used, you need to indicate the supplier, city and Country.

Line 159, what does 22,500 mean?

The photographs you have taken are not the best way of showing the roughness. I wouldn't include them.

You should add a paragraph on the conclusions.

Something weird happened with the citations. They are highlighted of different colours.

Comments on the Quality of English Language

Minor English editing

Author Response

Thank you for reviewing my paper despite your busy schedule.

We have provided answers to your questions in the attached file.
thank you

Reviewer 2 Report

Comments and Suggestions for Authors

Low temperature of Ag paste is the most conductive material for metallic powders, which can be wildly used in electrodes for flexible substrates. This manuscript focused on the physical properties of Ag paste derived from WP or DP route. The manuscript characterized the SEM, BET, Viscosity, and Resistivity, concluding that DP Ag particles are better suited than WP Ag particles for applications involving low-temperature curing Ag pastes. All the results and discussion are helpful for the readers working on optoelectronic functional material or devices. However, the authors should pay attention to the following questions:

1.     This manuscript provides many comparative studies, but the design and innovation of the paper are slightly lacking. I suggest the author add the importance (parameters related to this manuscript) and application cases of conductive Ag paste in the introduction section at least.

2.     The manuscript should include each descriptions to all the figure captions. Eg., There are not descriptions on “Table 1” in the manuscript; no Figure 2 (a) (b)…

3.     Section 3.3. The author claim that the conductivity of Ag sheets is completely different, which is resulted from the carbon content (one is 0.01, and the other is 0.3wt%.). Here, we suggested the author check this result, firstly the Carbon is conductive; secondly will the binder (containing C-content 0.3wt.% ) affect the conductivity?

 I suggested the manuscript can be accepted after minor revisions.

Comments on the Quality of English Language

None

Author Response

(The authors gave the same response as above.)

Reviewer 3 Report

Comments and Suggestions for Authors

This study compares the characteristics and low-temperature curing properties of pastes prepared from silver powders synthesized by either wet powder (WP) or dry powder (DP). The authors provide a variety of characterization methods to demonstrate the physical property differences between DP and WP. The comparison was intriguing yet I found it failed to provide strong arguments for their conclusion. TGA of DP and WP didn’t show obvious weight differences, yet the conductivity was dramatically different. I suggest providing more solid evidence for the conclusions if accepted.

1.      BET figures and data of both DP and WP should be included. Please explain why BET can indicate the size differences of Ag particles. FTIR compared the organic adsorption peaks between DP and WP. However, it is not indicated how the background was measured, I found this comparison insufficient to conclude.

2.      The argument for high electric conductivity is only about the carbon residual % differences between DP and WP. Have you compared using different treatments to remove more carbon from WP to see if the electricity conductivity increases?

3.      It is a bit confusing why there is plenty of discussion on the microstructure of DP and WP, but the argument of why DP is better than WP is around the residual carbon % of WP. The logic is rather loose. 

Author Response

(The authors gave the same response as above.)

Round 2

Reviewer 1 Report

Comments and Suggestions for Authors

The authors have answered my questions.

Comments on the Quality of English Language

Minor revisions

Reviewer 3 Report

Comments and Suggestions for Authors

The authors provided sufficient information following the previous comments and I think it can be accepted in current form.